# Penetration Overload Prediction Method Based on a Deep Neural Network with Multiple Inputs

**Haoran Ma, Hang Sun and Changsheng Li \***

The Department of Mechanical Engineering, Nanjing University of Technology and Science, Nanjing 210094, China
* Correspondence: lichangsheng1984@163.com

**Abstract:** In the process of high-speed penetration, penetrating ammunition is prone to problems such as penetration overload signal vibration and mixings and projectile attitude deflection. It is easy to misjudge if a fuze relies only on the overload data from the ground or the utilized program, and the actual penetration overload measured under actual launch conditions cannot be taken as the dynamic judgement basis. Therefore, a real-time penetration overload prediction method based on a deep neural network is proposed, which can predict overload values according to the projectile parameter settings, the real-time collection of overload information, and the calculation speed and assist the fuze in judging the target layer and projectile attitude. In this paper, we adopt a deep learning model with multiple time series inputs and modify the input coding mode so that the model can output a 48 μs overload curve within 20 μs, meeting the real-time signal processing requirements of the high-speed missile penetration process. The mean squared error between the predicted curve and the actual curve is 0.221 for the prediction of multilayer penetrating targets and 0.452 for the prediction of thick penetrating targets. A penetration overload prediction function can be realized.

**Keywords:** fuze penetration; hard target penetration; machine learning; overload prediction

## 1. Introduction

With the intensification of battlefield confrontation situations, the functional requirements of penetration ammunition must be improved to efficiently damage targets. The process of a projectile penetrating a concrete target at a high speed is very complex; it includes the friction effects of materials under high temperature and pressure conditions, the crack propagation of concrete, the strain rate effect and the stress wave propagation process [1]. The main problems of layer count initiation control include the following: (1) as the speed of a penetrating warhead increases, the fuze bears a higher impact overload, and the interlayer overload changes dramatically at the same time; (2) during the process of high-speed penetration, the vibration of the projectile body leads to the mixing of the penetration overload during the warhead penetration process [2]; and (3) sufficient adaptive detonation control cannot be achieved for targets with different hardness levels, thicknesses, falling speeds and attack angles [3]. The main problems concerning thick target initiation control are (1) the deflection of the missile body attitude, which leads to initiation control failure [4], and (2) the irregular overload changes caused by the uneven densities and material differences of the thick target penetrating the target medium [5]. The above problems make it difficult for a fuze to accurately identify the target layer under the layer count initiation control mode and make the accuracy of layer count initiation decrease or even cause the process to fail. Under the thick target initiation control mode, the attitude deflection of the projectile cannot be detected in real time, and the penetration state of the thick target cannot be judged normally through the real-time penetration overload.

With the development of modern computer technology, the application of artificial intelligence in the military field is exhibiting a rapidly rising trend. At the same time, in

the fields of target recognition, trajectory correction, etc., there are cases in which artificial intelligence technology is superior to traditional algorithms. In machine learning, to solve the problems of time series prediction and classification, many deep learning-based neural networks such as long short-time memory (LSTM) networks [6] and gate recurrent units (GRUs) [7] have been proposed. These models are called traditional cyclic neural networks [8]. Google proposed the transformer architecture [9] in 2017 to solve the problem of translating between different languages. This model has good performance in time series prediction problems.

## 2. Related Work

In the past, domestic and foreign scholars used traditional mechanical filtering or original signal filtering methods according to a single cut-off frequency [10,11] to filter penetration signals. Such methods lack effective theoretical support and have difficulty meeting the requirements of high accuracy. Reference [12] used singular value decomposition to reduce the signal noise, and the reconstructed penetration overload curve could better reflect the penetration process. Reference [13] was able to better decompose multichannel mixed signals by using underdetermined blind source decomposition and feature extraction. In reference [14], a multilayer perceptron (MLP) was used to predict the number of penetration layers, but without considering the impact beyond the time sequence, the losses of training set and test set were large. References [15–18] performed signal preprocessing on experimental data and simulation data based on empirical mode decomposition [19] and variable mode decomposition [20], respectively, which can clearly decompose the intrinsic mode functions (IMFs). References [21,22] carried out box-filter processing on the original penetration signal and dynamic threshold algorithm processing on the signal envelope. A new identification method was proposed in reference [23]; this approach uses a short-time Fourier transform to filter multilayer penetration signals and obtain layer number information according to the signal frequency components. References [24,25] conducted filtering analysis on the penetration overload signals of different types of warheads, such as flat-nosed projectiles and earth-penetrating projectiles, to study their internal mechanisms. References [26,27] studied the penetration overload mechanism at high and low penetration speeds.

## 3. Method

Aiming at the detection inaccuracy of traditional overload sensors and the detonation control of penetrating multilayer targets and thick targets, this paper improves the Informer model based on the transformer architecture [28] and names the resulting approach Penetration Fuze Informer (PF-Informer) to predict penetration overload curves. Through the machine learning of simulation data and experimental data, the mean squared error (MSE) and mean absolute error (MAE) are used to evaluate the model, and good results are obtained in a subsequent algorithm verification. In this paper, the short time series prediction problem for a single parameter that only depends on time as the input is creatively transformed into a long time series prediction problem with multivariate parameters. By recoding the time code and optimizing the constant parameter input, the calculation speed of the model is accelerated, the accuracy of the model is improved, and the timeliness and high accuracy requirements of the penetration fuze are met. The model has a stronger time series prediction ability than other models and can predict complete penetration overload curves under different penetration conditions, providing it with a broad application background.

### 3.1. Penetration Simulation and Data Set Introduction

3.1.1. Selection of the Penetration Overload Data Extraction Position

In this paper, a deep learning model is used to integrate the parameters of penetration overload and penetrating ammunition to train on existing simulation data and experimental data. The parameters related to penetration ammunition, such as the weight, length and

angle of attack, are set in advance. We combine these parameters with the overload curve of the short-term target impact and the calculated speed and incorporate them into the prediction model. The model transfers the predicted penetration overload curve to the fuze as a priori detonation control data. A data set is obtained through simulation and live ammunition experiments. In the simulation model, the bullet body and its bottom bolt are made of high-strength alloy steel (30CrMnSiNi2A) and the * MAT_ PLASTIC_ KINEMATIC constitutive model. This model can adopt the mixed hardening mode or the isotropic hardening mode. The isotropic hardening mode is adopted in this paper. The strain rate effect of this model is expressed by the Cowper–Symonds model. During the high-speed penetration process, the concrete is under large strain, high pressure and a high strain rate, so * MAT is adopted in this paper with the _ JOHNSON_ HOLMQUI ST_ CONCRETE constitutive model, or HJC constitutive model for short. The overload curve of the fuze simulation and test is greatly affected by vibration, and the oscillation and aliasing phenomena are serious. The penetration overload data at the missile body are selected as the data set. Figure 1 shows the simulated penetration overload curve extracted from the fuze, and Figure 2 shows the simulated penetration overload curve extracted from the missile body, where the sampling interval is 20 μs. In Figure 1, the overload signals extracted from the fuze end oscillate and mix severely, and it is impossible to distinguish the overload change during penetration. The overload curve extracted from the missile body in Figure 2 obviously changes, which can clearly reflect the penetration process.

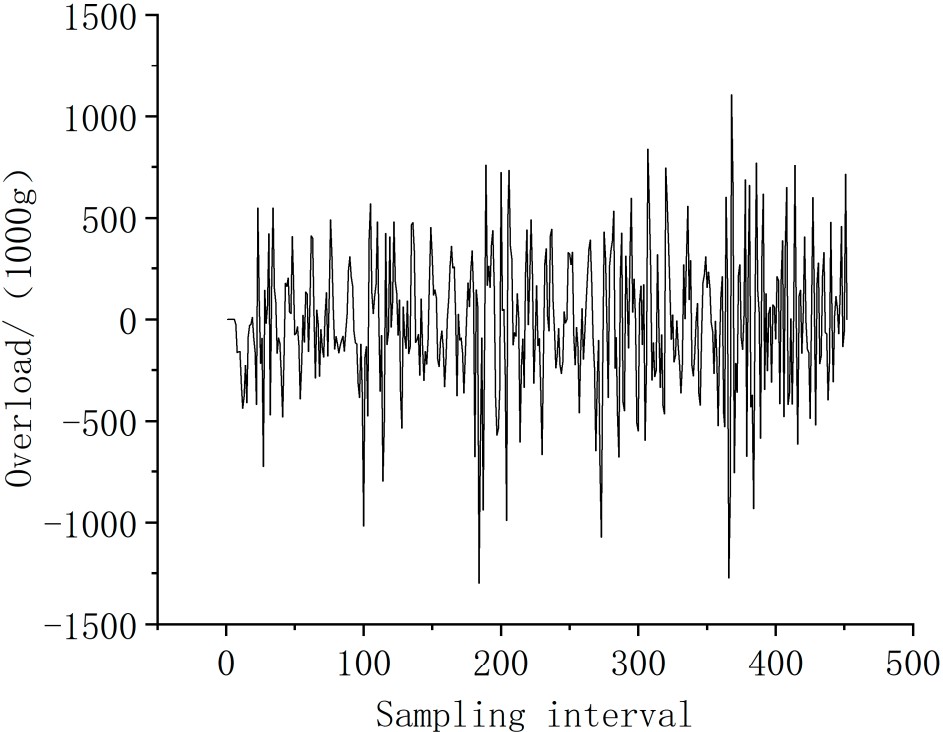

**Figure 1.** Simulation overload curve of multilayer target fuze terminal.

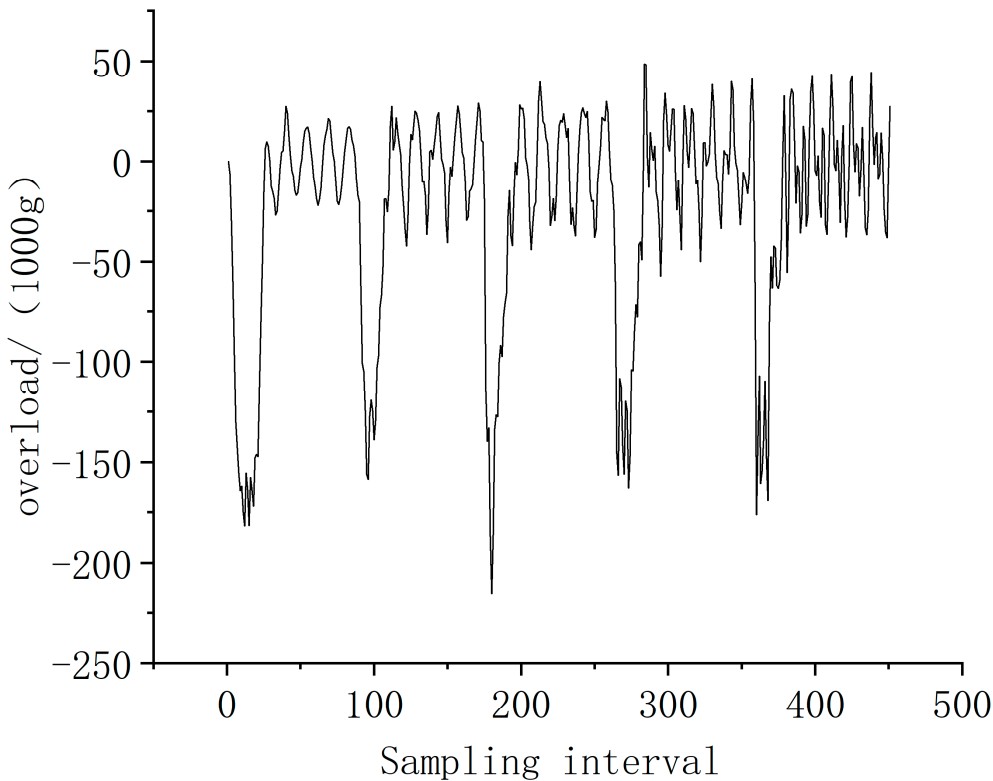

**Figure 2.** Overload curve of the multilayer target missile body simulation.

3.1.2. Selection of the Penetration Overload Data Set Label Type

The data set is composed of simulation data and experimental data. The data set is divided into a training set, verification set, and test set according to a certain ratio to ensure mutual exclusion between the training set, verification set, and test set; this is commonly known as the "reserve method". The reserve method usually divides large data-sets into sections of 90% for training, 5% for verification and 5% for testing, and small data sets are divided into 60%, 20% and 20% proportions. Due to the limited size of the data set, the data set is divided into a 60% training set, a 20% verification set and a 20% test set. In Figure 3, the data set labels of multivariate time series prediction are composed of time, bullet length, bullet weight, speed, angle of attack, and overload information from left to right. Overload is the target data category to be predicted. Furthermore, the data set contains overload data for multilayer targets and thick targets.

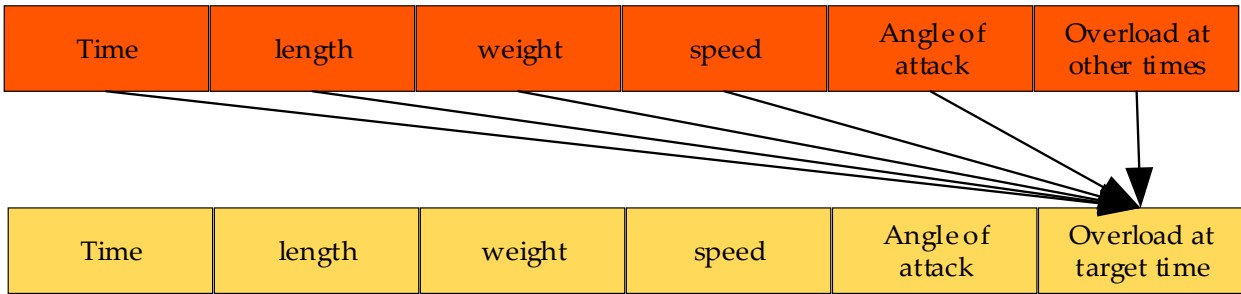

**Figure 3.** Overload curve with the multilayer target missile body simulation.

*3.2. Model Introduction*

A transformer has a strong short sequence prediction capability, but it has the disadvantages of high memory and high complexity due to its own architecture problems when it performs long-term time series prediction. We cannot directly incorporate the transformer's encoder–decoder system into the time series prediction problem. Informer is

an improved long-time sequence prediction model based on the transformer model, which uses both the encoder and decoder of a transformer.

### 3.2.1. The Principle of the Transformer

For the transformer model, the traditional convolutional neural network (CNN) and recurrent neural network (RNN) are abandoned, and the whole network is composed of a new mechanism named attention. The transformer block is composed of attention and a feedforward neural network. A neural network based on a transformer can be built by stacking transformer blocks. The transformer model utilizes parallel computing logic whose input and output for each point are independent. The calculation process at any time is based on all time inputs and the time coding of the corresponding inputs. A CNN can only select local inputs for calculation purposes. When using a CNN to make time series predictions, a serious issue is encountered because it cannot smoothly connect the relationship between the two convolution boxes. The prediction curve jitters from the perspective of global time series prediction. RNNs exhibit serial computing logic. The limitation of the RNN calculation process is that the calculation step for the current time $t$ is based on the calculation results from time $t - 1$, which greatly limits the calculation speed of the model. At the same time, an RNN loses the information contained in early calculation results when calculating long time series.

A transformer conducts model training and prediction through its encoder–decoder system. For each encoder, the data first pass through the self-attention module to obtain an attention matrix containing three inputs (query, key, value), and then the output probability value is obtained by activating the softmax function:

$$A(q_i, k_i, v_i) = Softmax\left(\frac{QK^T}{\sqrt{d}}\right) V \tag{1}$$

The training logic is as follows: $Q$ is the query matrix, $K$ is the key matrix, $V$ is the value matrix, $I$ is the matrix of inputs $X_i$, R is the normalized matrix, $K^T$ is the transposed matrix of $K$, $A'$ is the matrix obtained after we normalize matrix $A$, $O$ is the matrix of outputs $B_i$, and $W^q W^k W^v$ are the parameters matrix that can be learned by the model.

$$\begin{aligned} Q &= W^q * I \\ K &= W^k * I \\ V &= W^v * I \\ A &= K^T * Q \\ A' &= R * A \\ O &= V * A' \end{aligned} \tag{2}$$

For the attention value corresponding to the $i$th query, the following probabilities can be used to substitute Equation (2) into Equation (1):

$$A(q_i, k_i, v_i) = \sum_j \frac{k(q_i, k_j)}{\sum_l k(q_i, k_l)} v_j = E_p(k_j | q_i)[v_j] \tag{3}$$

where $k(q_i, k_l)$ is the asymmetry index core in the attention mechanism, which is utilized as $e^{\left(\frac{q_i k_j^T}{\sqrt{d}}\right)}$. The attention mechanism describes the similarity between queries and keys and aims to select important q-k pairs.

### 3.2.2. The Improved PF-Informer Model
Informer Model Architecture

The Informer model is shown in Figure 4.

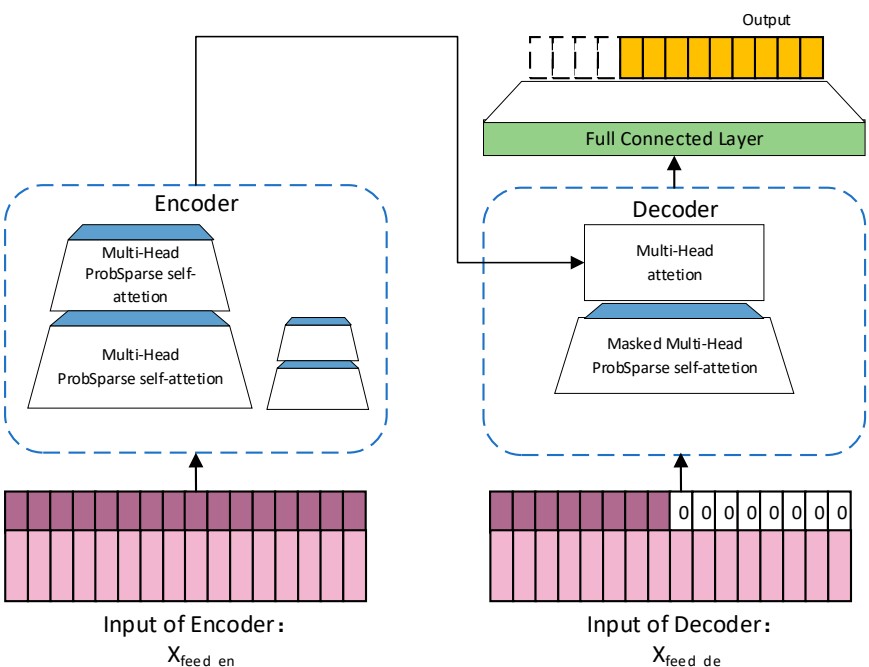

**Figure 4.** The structure of the Informer model.

Informer uses the Kullback–Leibler divergence measure [29] (commonly known as KL divergence) to measure the attention similarity. The smaller the metric value is, the closer the two tested distributions are. After simplification, the sparsity measurement formula for the *i*th query is defined as

$$M(q_i, k) = In\sum_{j=1}^{L_k} e^{\left(\frac{q_i k_j^T}{\sqrt{d}}\right)} - \frac{1}{L_k}\sum_{j=1}^{L_k} \frac{q_i k_j^T}{\sqrt{d}} \tag{4}$$

The first half is the Log-Sum-Exp (LSE) of $q_i$ for all keys. The second half is the arithmetic average of $q_i$ for all keys. According to the above evaluation method, the ProbSparse self-attention formula is obtained, where $\overline{Q}$ is the sparse version of the original $Q$ matrix. The Informer model uses ProbSparse self-attention to select appropriate q-k pairs.

$$A(q_i, k_i, v_i) = Soft\max\left(\frac{\overline{Q}K^T}{\sqrt{d}}\right)V \tag{5}$$

To efficiently solve the time series prediction problem, the encoder inputs multivariate long time series data and replaces the multihead ProbSparse self-attention mechanism with a multihead self-attention mechanism to extract multiple important pieces of attention. The decoder module also uses multihead ProbSparse self-attention to improve the robustness of the algorithm. The multihead ProbSparse self-attention greatly reduces the computational complexity, from O (L ˆ 2) to O (LlogL), in terms of time complexity and memory usage (L is the time sequence length), making the model capable of processing long time series. The ProbSparse self-attention used in the model optimizes the transformer's self-attention mechanism, calculates the upper sparse evaluation boundary, and optimizes the computational time and spatial complexity of the algorithm. The Informer solves the computational complexity of the transformer through this method.

In the encoder part, a one-dimensional convolution operation is used on the time series extracted by the multihead ProbSparse self-attention, and an exponential linear unit (ELU) is used as the activation function. The model uses the maximum pooling operation at the end. Compared with the rectified linear unit (ReLU) activation function, the ELU has a negative value, which makes the average activation value closer to zero. The closer the average activation value is to zero, the closer the gradient is to the natural gradient, which

speeds up the learning process of the model. As an unsaturated function, the gradient of the ELU is nonzero for all its negative values, so no gradient disappearance or neuronal death problems are encountered. Neuronal death means that when an abnormal input occurs, a large number of gradients are generated in the back-propagation process, which causes neurons to die and the gradient to disappear. Compared with the ReLU function and its variants, the ELU can shorten the time required by the deep neural network training process and improve its accuracy [30]. The Informer solves the high memory usage of the transformer through this method.

$$f(x) = \begin{cases} x, \text{ if } x \geq 0 \\ \alpha(e^x - 1), \text{if } x < 0 \end{cases} \tag{6}$$

The parallel processing ability of the model improves the speed of long sequence prediction and reasoning and can satisfy the demand of predicting the subsequent short-term overload curve of penetration. When verifying the prediction accuracy, the decoder receives a part of the long input, covers the part of the input to be predicted at the tail, fills the value of the corresponding position of its timing to 0, generates a prediction and compares the result to the actual data to calculate the loss.

Modification of the Time Input of the PF-Informer Model

In the time series prediction task, time, as the main parameter input, is still very important. In the original Informer, an open-source library called "pandas" is directly used to extract and normalize the characteristic values (year–month–day–hour). This approach makes the average characteristic value 0, the maximum characteristic value 0.5 and the minimum characteristic value $-0.5$.

The two data sets contain the data of multilayer targets and thick targets. The numbers of sampling points acquired for the two types of targets are different. For multilayer targets, the first objective is to distinguish the target layers in the time sequence, and the priority is the slope of the overload change. Furthermore, the magnitude of the amplitude loss has little influence on distinguishing the target layers. According to different data sizes, two coding methods can be chosen: when the test set possesses a certain size, 1 to the end of corresponding sequence can be used for the time coding; in the absence of data set support, time coding can produce a sequence-length cycle consisting of 1 to a single-layer target. For thick targets, axial overload changes are very important. During the process of penetrating a thick target, the axial overload drops sharply due to the change in attitude, so the time coding process needs to cycle from 1 to the whole long sequence. The time sequence of the penetration process cannot be directly time coded using dates. Here, the time sequence is divided, and the time corresponding to each data point in the experimental process is coded as (starting position, starting position plus total time sequence length). The normalization logic is as follows:

$$Output = \left( \frac{Input}{\text{Length of corresponding sequence}} - 0.5 \right) \tag{7}$$

In this way, the characteristic value of the input time code is 0, the maximum value is 0.5, the minimum value is $-0.5$, and the variance is 1. The regularization operation is performed at the input, which can effectively improve the training progress.

Modification of the Non-Variable Physical Quantity Inputs of the PF-Informer Model

The PF-Informer model optimizes a large number of repeated and immutable physical input parameters, significantly reducing the MSE of the prediction model and making the model more robust. The multivariate inputs used by PF-Informer for the time series prediction process include time, speed, angle of attack, bullet weight, bullet length and other parameters. The PF-Informer model can easily adjust the speed-related weight according to the speed change trend. Parameters such as the attack angle, bullet weight

and bullet length are input into the model in advance, and the corresponding parameters in the time series do not change. Through a large number of training analyses, it can be concluded that adding too many immutable parameters causes serious model overfitting, making the universality of the model seriously worse. As shown in Figure 5, the change in the MSE of the prediction model is modified by adding categories that introduce constant input parameters.

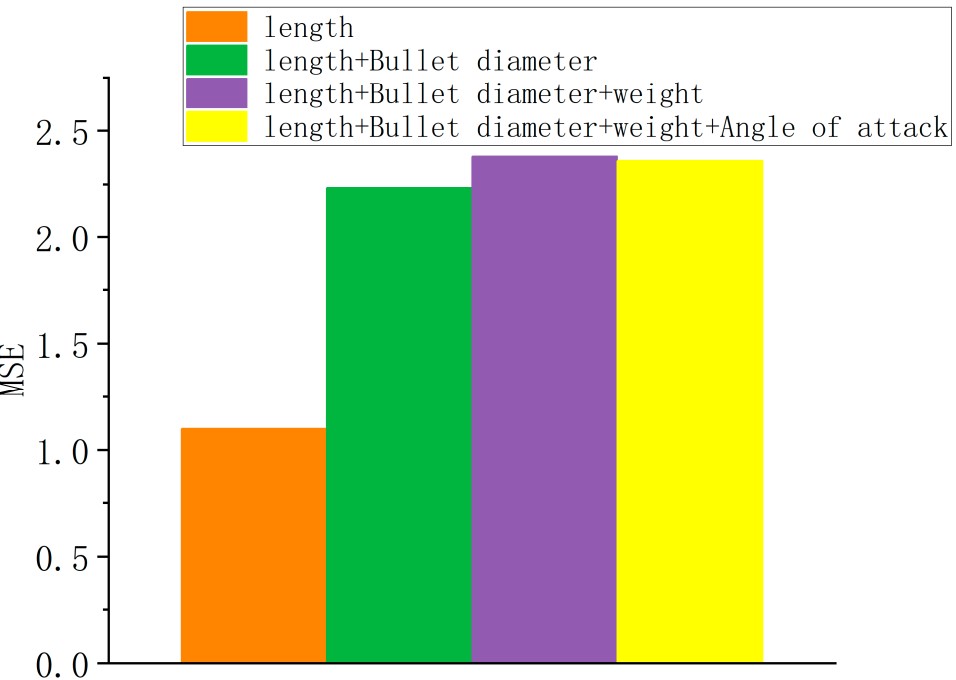

**Figure 5.** MSE diagram.

It is obvious from the Figure 5 that the introduction of more immutable input parameters significantly increases the MSE and decreases the prediction ability of the model. Since the model based on the transformer architecture inputs too many identical parameters, the self-attention weights corresponding to these invariable parameters are greatly increased, which results in the self-attention weights corresponding to other variable input parameters being reduced or even ignored during the model training process. Finally, the model deteriorates.

To prevent model weight disorder, the input parameters should be preprocessed. For a time series prediction task with a long sequence input, the model with the transformer architecture sends the characteristic values of each time series to the activation function. When multiple identical constant parameters appear, the value obtained by multiplying the query value corresponding to each key value is higher, which makes the corresponding weight obtained after being sent to the activation function abnormal. Here, we need to optimize the weight of the input of the entire constant. We need to change it to a single effective weight, and the weight of the rest of the series should be as small as possible. Here, the input softmax activation function is used to adjust the weight, as shown in Equation (8). The feature of this activation function is that its value at negative infinity is 0.

$$f(x) = \frac{1}{1 + e^{-x}} \tag{8}$$

The way to address invariant parameter input sequences is to add negative infinity to the input values at other positions in addition to the first valid input value of each sequence. In specific operations, a very large negative value is added to replace these values so that the value at the position with the maximum negative value is weighted to 0 after passing through the softmax activation function. Then, the weight of the exceptions is ignored. The

specific PF-Informer method is shown in Figure 6 below. Here, an input with a of 6° attack angle is taken as an example.

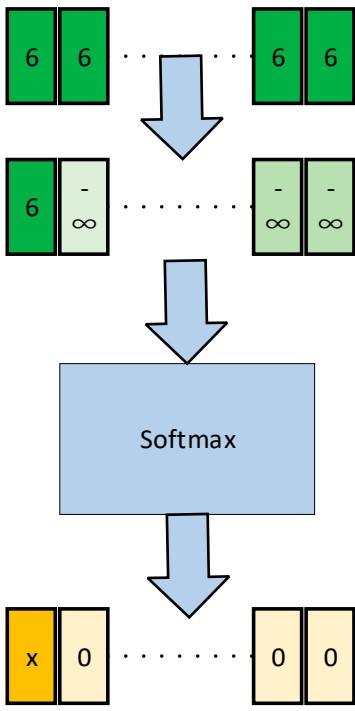

**Figure 6.** Input modification diagram.

Through the above data preprocessing technique, the normal input weight x corresponding to the constant parameter can still be obtained, and the weights corresponding to the other identical and redundant constant parameters can be set to zero, thus the weight disorder problem caused by too many identical parameters does not affect the entire model. With the introduction of constant parameters, the model achieves stronger universality and a better prediction ability. The change in the MSE is shown in Figure 7.

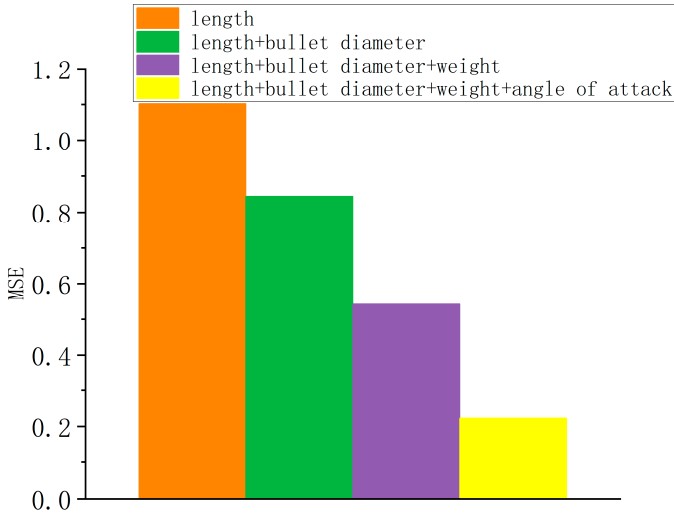

**Figure 7.** MSE change induced after input optimization.

## 4. Experiments

The experiment predicts the penetration overload curve through the deep learning network, and trains by mixing the simulation data and real data of the two targets as the data set. In order to verify that the accuracy of the time series prediction model with

multiple parameter inputs is better than that of the time series prediction model with single parameter inputs, the prediction models are compared by MAE and MSE based on data sets with the same capacity and type.

The experimental design is as follows:

(1)　The PF Informer model is used, in which time series prediction only takes time as input for model training.

(2)　The PF Informer model is used, in which the time and various projectile parameters are used as the input of time series prediction for model training.

After model training, in order to verify the generalization and reliability of the model, the model is verified by simulation experiments. Only the first 20 points of overload data are input to the model to simulate the overload data collected by the sensor in a short time when the actual target is hit. Then, the model is verified in two steps. The first step is to extract short time series from the validation set for validation, and explain the importance of multivariate parameter input to the model through MAE and MSE, and initially verify the feasibility of the model. The second step is to select two groups of simulation data and live ammunition data that do not exist in the training set, and input the first 20 points of simulation data and live ammunition data into the model to predict the long time series. By comparing the difference between the predicted simulation data overload and the actual simulation data overload, and comparing the difference between the predicted live ammunition data overload and the actual live ammunition data overload, the generalization and reliability of the model are further verified.

### 4.1. Training Environment

The experimental platform is Windows 11, and the Pytorch deep learning framework is used in Anaconda scientific computing platform. The version of Cuda core used is 11.7, the training model and the using model to predict the overload curve are based on the graphics processor NVIDIA GeForce RTX 3090ti. To ensure model universality and reliability, we need to consider many aspects: (1) the prediction ability of the model for different weight and length projectiles; (2) the prediction ability of the model for the same missile and different types of targets, such as multilayer and thick targets; (3) for the real live ammunition test, the vibration of the missile body is large, and there is a large gap between the amplitude and the simulation data. Whether the prediction loss of the model on the amplitude belongs to the acceptance category can further measure the prediction ability of the model's overload curve time series.

There are two types of samples: multilayer and thick target. The number of multilayer target samples is 200, and each sample has 450 sample points, forming a 90,000 sample points multilayer target overload data set. The number of thick target samples is 50, and the sampling points for each thick target are 450, forming a 22,500-point-thick target overload data set. The quality of prediction is measured by the loss, MSE and MAE on training set and test set.

### 4.2. Model Training

In the experiment, we use five encoders and five decoders to build the model. The dimension of the model is 512, and the number of attention heads is eight. To reduce the size of the model, the number of layers of the full convolutional network is set to 1024. The test is conducted twice each time, and the number of training epochs is 100. At the same time, the patience mechanism is called, and the patience value is set to 20. Patience mechanism means that when multiple training reaches a certain number of times, the test set loss does not decrease, and the training will be terminated in advance. The model is optimized using the Adam optimizer, which is not very sensitive to the size of the learning rate, so the learning rate is lowered to 0.0001, and the training set batch size is 128. The difference between the following experiments lies in the selection of multi-layer target data set and thick target data set, as shown in Table 1.

**Table 1.** Model hyper parameter settings.

| Encoder | Decoder | Model Dimension | Multiple Attention Heads | Number of Convolutional Network Layers | Training Epochs | Patience Training Rounds | Algorithm Optimizer | Learning Rate | Batch Size |
|---|---|---|---|---|---|---|---|---|---|
| 5 | 5 | 512 | 8 | 1024 | 100 | 20 | Adam | 0.0001 | 128 |

The loss of model training is as follows: the loss on training set is 0.00735071, the loss on verification set is 0.04294431, and the loss on test set is 0.03853120. When the model training reached the 42nd epoch, the loss of the test set has not decreased for 20 consecutive epochs and the model stopped training in advance.

### 4.2.1. Local Prediction Effect of the Multilayer Target Model

The local prediction results of the multilayer target model obtained through training are shown in Figure 8 below, where the overload value is normalized. The normalized data is used to verify the predicted performance of the model on the verification set. Figure 8a,b refers to the prediction data and the real data inputs obtained using only time, respectively, and Figure 8c,d refer to the prediction data and real data inputs obtained using multivariate elements. It can be clearly determined that the effect of multitime input prediction is better than that of prediction using only time as the input.

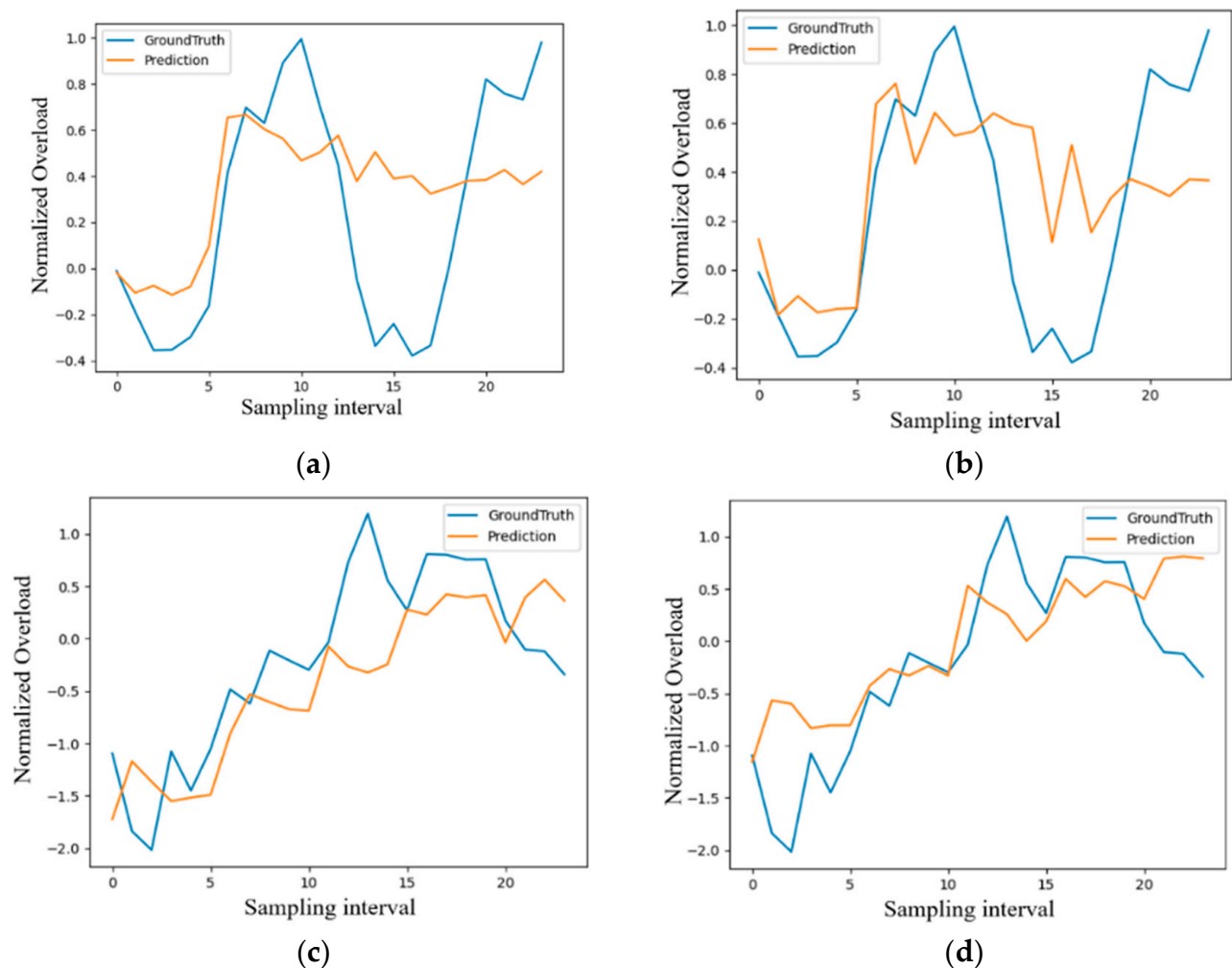

**Figure 8.** Local time series prediction for multilayer target. (**a**) First single timing prediction; (**b**) second single time series prediction; (**c**) first multivariate element prediction; (**d**) second multivariate element prediction.

### 4.2.2. Local Prediction Effect of the Thick Target Model

The local prediction result obtained by the thick target model through training is shown in Figure 9, where the overload value is normalized. Figure 9a,b refers to the prediction data and real data that only use time as the input, respectively, and Figure 9c,d refers to the prediction data and real data that use multivariate elements as inputs. It can be clearly determined that the effect of prediction with multivariate time inputs is far better than that of prediction using only time as the input, but the overall amplitude exhibits a certain deviation.

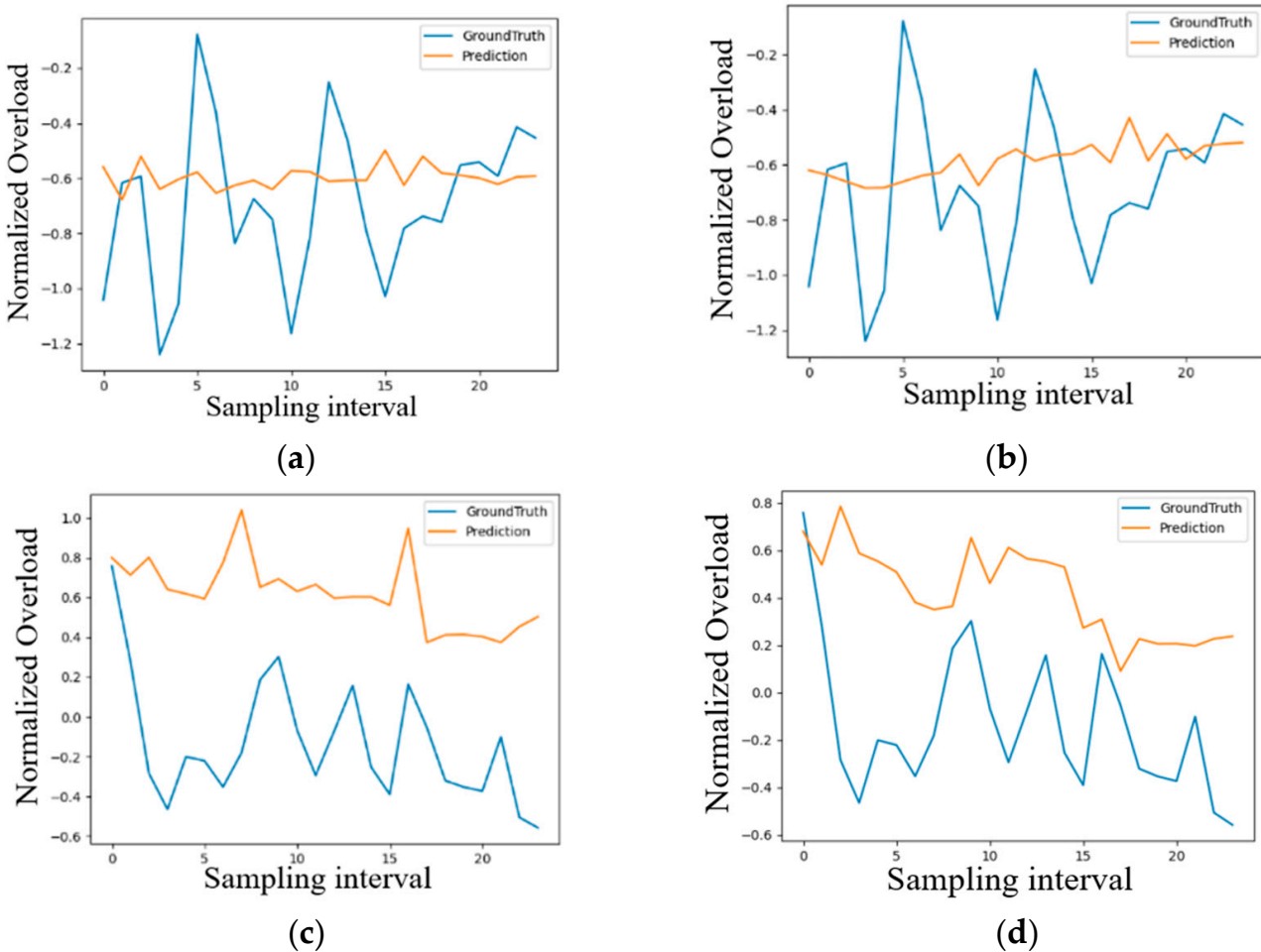

**Figure 9.** Local time series prediction effect obtained for thick targets. (**a**) First single timing prediction; (**b**) second single time series prediction; (**c**) first multivariate element prediction; (**d**) second multivariate element prediction.

Through the above verification on the verification set, the feasibility of the model is preliminarily verified.

### 4.3. Results of Model Prediction

The test uses the PF-Informer model to train the multilayer thick target models, and the single-time input and multi-element inputs are compared. The prediction effect is measured by the MSE and the MAE. The results of comparisons conducted in several prediction situations are shown in Table 2, and the visual data are shown in Figures 10 and 11.

**Table 2.** Evaluation of the model training results.

|  | Single Multilayer Target (First Test) | Single Multilayer Target (Second Test) | Multi Element Multilayer Target (First Test) | Multi Element Multilayer Target (Second Test) | Single Thickness Target (First Test) | Single Thickness Target (First Test) | Multi Element Thick Target (First Test) | Multi Element Thick Target (Second Test) |
|---|---|---|---|---|---|---|---|---|
| MSE | 0.411 | 0.421 | 0.226 | 0.221 | 5.395 | 5.560 | 0.462 | 0.452 |
| MAE | 0.491 | 0.502 | 0.260 | 0.252 | 1.825 | 1.856 | 0.718 | 0.705 |

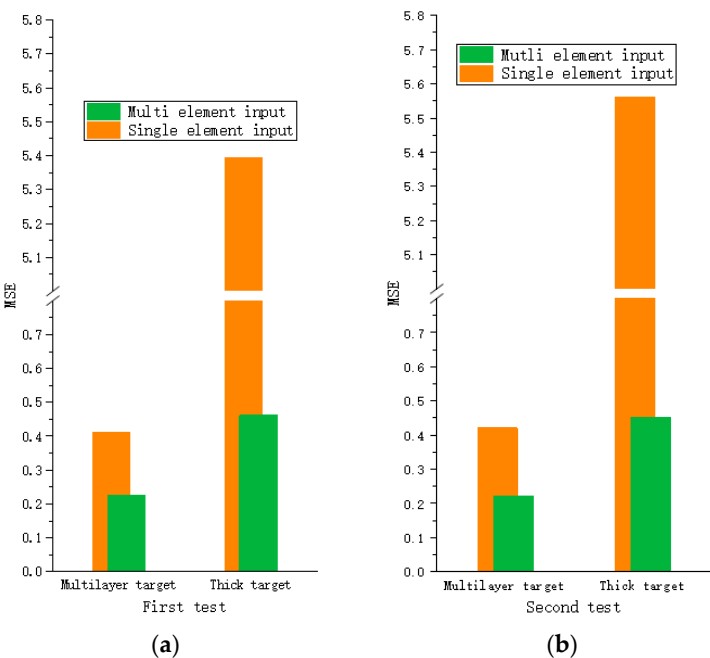

**Figure 10.** MSE performance achieved on the test set. (**a**) First test; (**b**) second test.

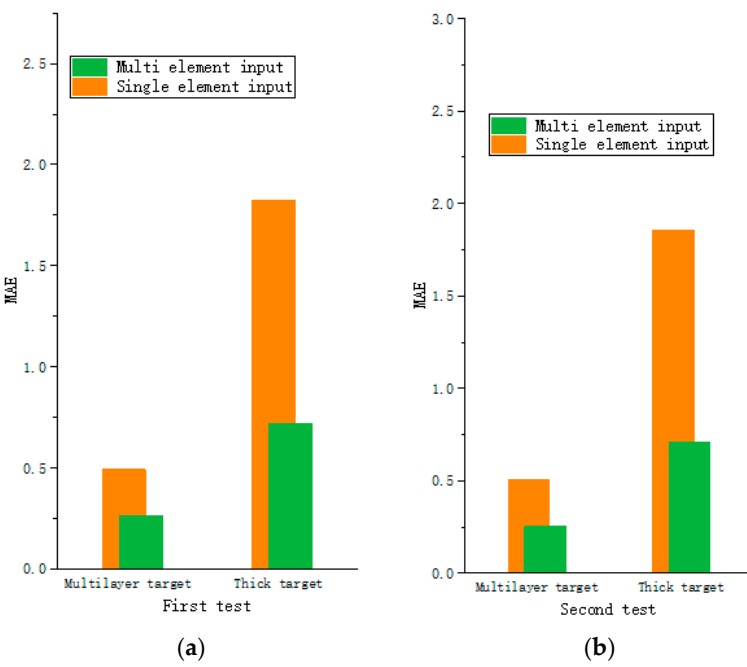

**Figure 11.** MAE performance achieved on the test set. (**a**) First test; (**b**) second test.

By analyzing the data in Table 2 and the figure, we can see that the prediction accuracy of the multivariate input is far better than that of the time-only input. Slight optimization is achieved for multilayer targets, but the effect is remarkable on multivariate thick targets, which demonstrates the importance of multivariate parameters in long time series prediction.

### 4.4. Forecasting Model Test

The prediction ability of the developed model is measured by comparing the data that do not belong to the model training set with the data predicted by the model. The overload and speed values of the first 24 sampling points are input into the model combing with the attack angle, bullet weight, bullet diameter, bullet length, etc., and the output is used as the input of the next time point to predict the subsequent overload curve. The comparison between the predicted overload curve for multilayer and thick targets and the real overload curve produced for the test set is shown in Figures 12 and 13.

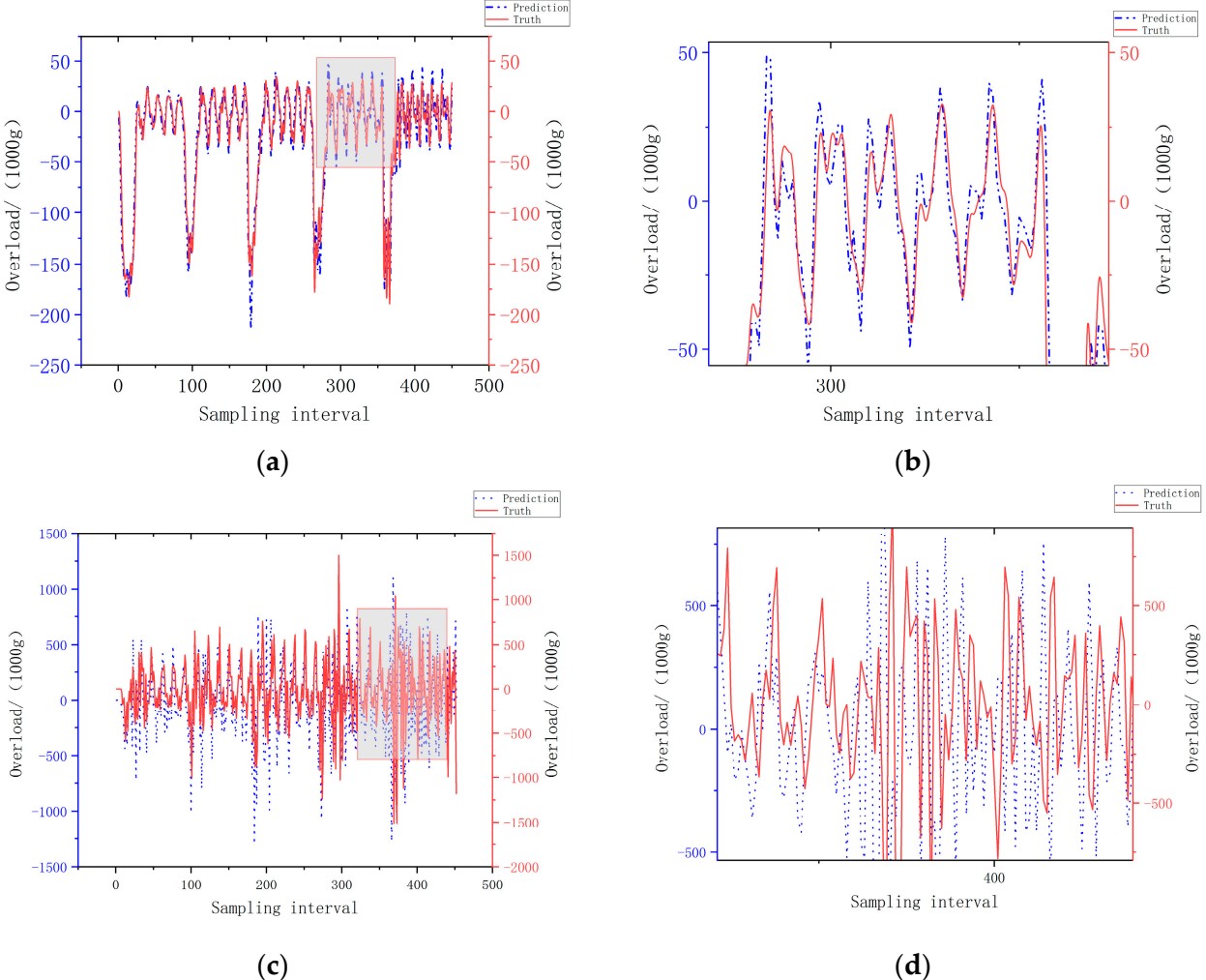

**Figure 12.** Prediction and actual overload curves for a multilayer target. (**a**) Global map of the multilayer target prediction results; (**b**) partial enlarged view of the multilayer target prediction results; (**c**) global map of the multilayer target prediction results; (**d**) partial enlarged view of the multilayer target prediction results.

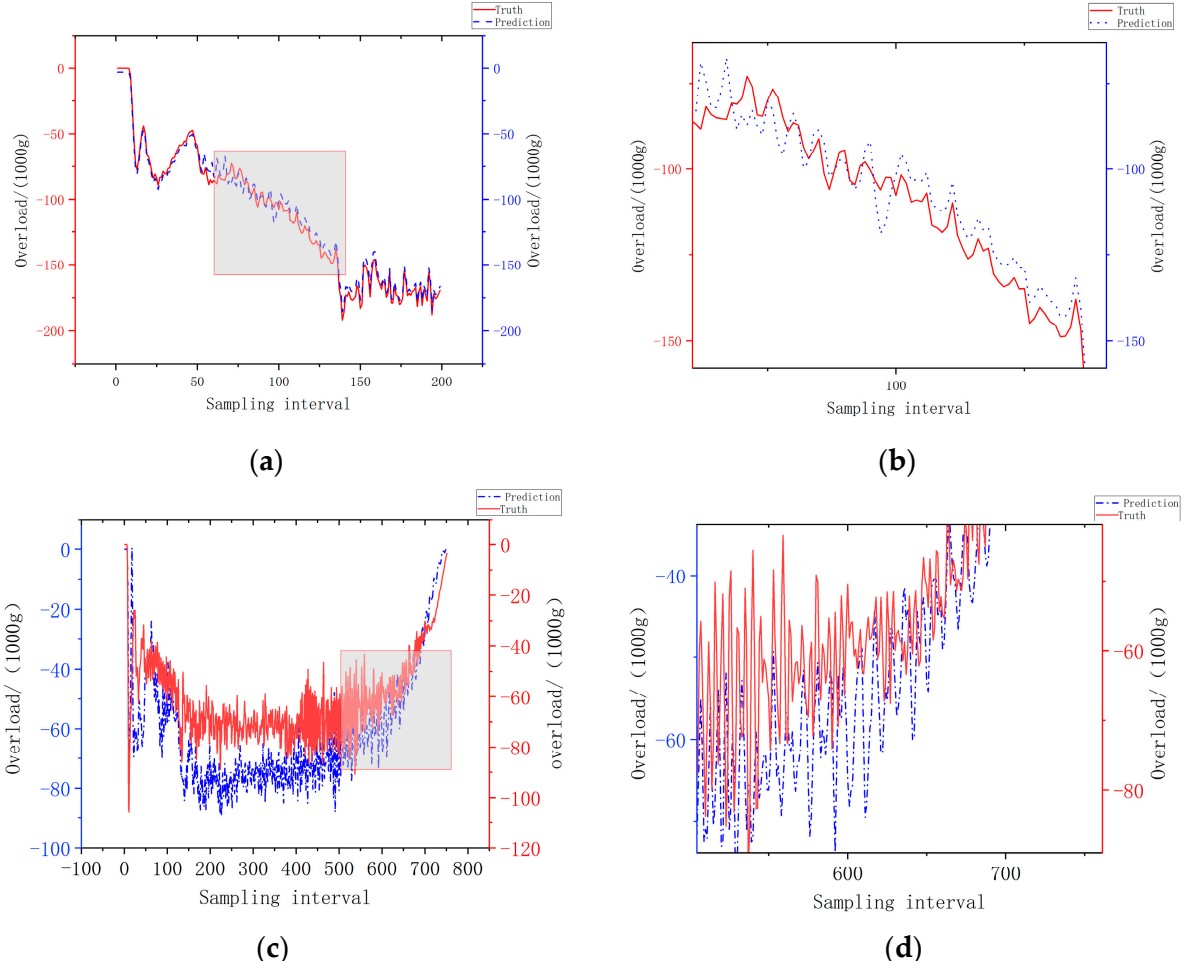

**Figure 13.** Prediction and actual overload curve for a thick target. (**a**) Global map of thick the target prediction results; (**b**) partial enlarged view of the thick target prediction results; (**c**) global map of thick the target prediction results; (**d**) partial enlarged view of the thick target prediction results.

Figure 12a,b are predictions using simulation data, and Figure 12c,d are predictions using live ammunition data. It can be seen from the global and local magnified prediction graphs in Figure 12a,b that the overload prediction for multilayer targets is highly consistent, and there is a slight deviation in the amplitude prediction for the peak overload value, which can meet the demand of the overload curve prediction. It can be seen from Figure 12c,d that although the prediction based on live ammunition data has some deviation in amplitude, the basic process of target penetration can be consistent. Figure 13a,b are predictions using simulation data, and Figure 13c,d are predictions using live ammunition data. It can be seen from Figure 13a,b that the prediction effect based on simulation data is basically consistent, with only a slight deviation in amplitude. It can be seen from Figure 13c,d, the global prediction map and the local zoomed in region, respectively, that the overload prediction trend for thick targets is basically consistent, but the prediction gap between specific amplitudes is obvious. The large gap between multilayer and thick targets is mainly due to the periodic changes in the multilayer target data, the obvious changes in the multilayer target overload, and the fact that the model's attention mechanism easily focuses on the locations with obvious changes. In contrast, the thick target data span is long, the change trend is mild, and the model attention mechanism has difficulty focusing, leading to inaccurate prediction results. At the same time, the gap between the prediction based on simulation data and the prediction based on live ammunition data is that the training set is dominated by simulation data, so the prediction effect for simulation data is better.

### 4.5. Comparison with Other Models

In addition, we also compare other models with PF-Informer, and the experimental results are shown in the following table. The data set used is multilayer target data set, and the comparison models are GRU-ODE-BAYES, GRU, LSTM, the autoregressive integrated moving average (ARIMA), a CNN, and N-BEATS. The results are shown in Table 3.

**Table 3.** Training results of different models.

|  | PF-Informer | GRU-ODE-BAYES | GRU | LSTM | ARIMA | CNN | N-BEATS |
|---|---|---|---|---|---|---|---|
| MSE | 0.221 | 0.366 | 0.464 | 0.452 | 0.932 | 0.822 | 0.343 |
| MAE | 0.252 | 0.387 | 0.520 | 0.502 | 1.012 | 0.942 | 0.369 |

We determine that PF-Informer performs best among the tested models, and its MSE and MAE are very low.

## 5. Discussion

In this section, we discuss the training of Transformer model based on small batch data, and the calculation accuracy and speed of the model. This paper introduces a better time series prediction model based on limited data. At the same time, the future research directions and difficulties are considered.

### 5.1. Model Training of Small Batch Data

We must admit that a large amount of high-quality data is extremely important for training the deep learning network, especially for Transformer. This model will be far ahead of other models in large data sets, but in small data sets, it will introduce the problem of over-fitting that is difficult to control [31,32]. The first thing to consider is diversification of the data, which greatly increases the dimension of the data, so that the unique attention mechanism of the model can better pay attention to the connection between multiple data rather than the connection in a single time. Through this method, the over-fitting of the model is well suppressed.

### 5.2. Model Accuracy and Operation Speed

For Transformer, when the decoder is used for prediction, the subsequent overload curve is obtained by masking the subsequent input through the upper triangular mask matrix. This approach is similar to that of the traditional RNN network, but for the RNN network, it is difficult to correlate the input information at the current time with the input information at the early time. When using RNN to predict the time series of a long series, it is easy to forget the first half of the information, and Transformer will integrate all the data not covered by the mask matrix to make the prediction, effectively ensuring the accuracy of the model. Because the model uses KL divergence to calculate attention, the computational complexity is changed from $O(L^2)$ to $O(L\log L)$, which greatly reduces the computational difficulty. Time series prediction that can output 48 μs within 20 μs has been emphasized many times in this paper. First of all, from the perspective of sensor sampling, it is customary to take 2 0us as a time period to sample the overload value, so after obtaining the overload value of 20 μs, we hope that we can predict longer overload data through the model within 20 μs. All the calculations in this paper have been integrated and expanded into large matrix calculations. Take encoder as an example to estimate the number of model parameters. Encoder includes word embedding layer, multi-head attention layer, fully connected layer and layer normalization layer. According to the hyperparameter settings of the model, the number of parameters used in the word embedding layer is about 2,304,000, and the number of parameters used in the multi-head attention layer is 1,048,576. The formula of the fully connected layer is as follows:

$$\text{FFN}(x) = \max(0, xW_1 + b_1)W_1 + b_2 \tag{9}$$

The parameters of the fully connected layer are 2,097,152, the layer normalization layer is 17,408, and the total parameters of the encoder are 3,393,536. Each head in each encoder is parallel, and the Q, K, V matrices of each header are parallel, so most matrix calculations can be performed synchronously. The floating-point performance of the computing resources used is 40 TFLOPS. Therefore, it is possible to achieve the prediction of 48 μs or even longer, within 20 μs.

### 5.3. Motivation and Contribution of the Study

The mechanical and electromagnetic environment of the penetration fuze signal processing system is complex and harsh during the high-speed penetration of the projectile into the reinforced concrete target. In the past, experts in the field of penetration fuze mainly used the actual sensor signals to identify the flight status and control the detonation. In recent years, with the complex and changeable targets, the measured signals are more oscillatory, mixed and fuzzy, and the fuze initiation control is facing more severe challenges. With the development of technology, the chip processing ability has improved and new algorithms have emerged, and the penetration overload prediction has gradually attracted the attention of experts in the field. Through the use of dynamic prediction method, combined with real-time overload data, flight status identification and detonation control can be carried out. Even if the projectile body has a severe deflection, it can detonate after the appropriate delay time through this method to prevent the phenomenon of no explosion or early explosion, and improve the damage effect. Therefore, penetration overload prediction is of great significance in the field of penetration ammunition control, and is also one of the research focuses in this field this year.

### 5.4. Limitations and Future Direction

In some fields, some data are very difficult to obtain, and the data set is non-public. The data set used in this paper is based on years of simulation and live ammunition data, and the capacity of the data set is not large. If the model used in this paper is based on a larger and richer data set, it will further improve its accuracy, so the following work is aimed at collecting more simulation and live ammunition data of different warheads. The following research needs to realize the hardware implementation based on missile-borne resources. For the ammunition, the resources on the ammunition are very scarce. It is necessary to carry the model with weak hardware and low power consumption. Therefore, the subsequent research needs to balance the size of the model and the prediction accuracy, and realize the prediction function by carrying the model on FPGA.

## 6. Conclusions

In this paper, we use the PF-Informer model to predict the overload curves of multilayer and thick targets and solve the model overfitting problem caused by excessive time coding and invariant parameters. By recoding the time code, the time code at the time input is regularized, the input mode of the invariant parameters is optimized, the abnormal weights are removed while retaining the effective weights, and the model overfitting effect is reduced. PF-Informer uses an encoder–decoder system for characteristic value extraction, coding training, and model prediction, and this system is far superior to the traditional deep neural network. On the basis of sufficient independent computing units, the model can output 48 μs overload curve within 20 μs, and its prediction accuracy and speed can be guaranteed to a certain extent. From the verification of the algorithm, this method can adaptively predict the overload curves induced by the penetration process under different conditions.

**Author Contributions:** Methodology, H.M.; Software, H.S.; Validation, H.M.; Writing—original draft, H.M.; Supervision, C.L. All authors have read and agreed to the published version of the manuscript.

**Funding:** This research was supported by the Pre-research fund (9090102010304, 909030404, 90903040413).

**Institutional Review Board Statement:** Not applicable.

**Informed Consent Statement:** Not applicable.

**Data Availability Statement:** The data are not publicly available due to privacy.

**Conflicts of Interest:** The authors declare no conflict of interest.

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
