# Peer review of "Penetration Overload Prediction Method Based on a Deep Neural Network with Multiple Inputs"

_applsci, doi:10.3390/app13042351_

Round 1

Reviewer 1 Report

This paper is related  to Penetration Overload Prediction Method Based on a Deep Neural Network with Multiple Inputs. The predicted results have not validated by experimental results.  It is commended to validate the prediction results by comparing experimental results.

Reviewer 2 Report

(1) In general, this paper does not seem to carry out real experimental research, but uses simulation signals. From this point of view, I think it is not suitable for this journal. It is hoped that the author can increase the experimental research, and the experimental research should add the specific experimental process and details.

 (2) The main algorithm adopted by the author seems to have no obvious innovation in the field of signal processing. So I hope to carry out more in-depth research.

Reviewer 3 Report

1. The figures in the paper are vague and unclear,  and there are redundant lines in the figure 10 and 11.

2. The font size of some formulas in the paper is not uniform and needs to be modified.

3. Add the meaning and unit of horizontal and vertical coordinate axes in the figure 8 and 9.

4. "the model can output a 48-us overload curve within 20-us, meeting the real-time signal processing requirements of the high-speed missile penetration process."  is mentioned in the Abstract and Conclusion of the paper, but there is less discussion on the calculation time in the paper. It is suggested to supplement relevant contents.

Reviewer 4 Report

This manuscript proposed a modified penetration overload prediction method based on deep learning, the topic looks interesting. My comments are as follows:

1) Both motivations and contributions are unclear, please refine them.

2) Separate related work section is suggested to comprehensively review recent development.

3) High-quality figures are suggested to better demonstrate both the proposed method and experimental results.

4) More evaluation metrics and state-of-the-arts should be included in the experiments to make the experiments more sufficient.

5) Separate discussion section should be considered to discuss both limitations and future directions. For example, how to further enhance the performance of deep learning model by using the prior knowledge? How to accelerate the deep learning model training? Some related papers are recommended: physics-informed deep learning for musculoskeletal modelling: predicting muscle forces and joint kinematics from surface emg, IEEE TNSRE, and non-iterative and fast deep learning: multilayer extreme learning machines, JFI.

Round 2

Reviewer 1 Report

It is well written and is recommended for publication.

Author Response

Thank you for your affirmation. Your comments are of great significance to my article.

Reviewer 2 Report

The author claims to have carried out further experimental research, mainly including:

1. The real ammunition recovery data of two kinds of penetrating targets are sent into the model for prediction and comparison;

2. Analyze the difference between simulation data prediction and live ammunition recovery data prediction.

However, I compared the original and the revised version, and I did not find the experimental content.

If the experiment is carried out, it is necessary to describe the experimental process, experimental equipment, original experimental data, experimental data processing process and conclusions.

In addition, the quality of the pictures in the revised version is too poor and too vague.

Reviewer 4 Report

The quality of this paper has been improved, but some of my comments are still not addressed. For example, the quality of some figures are still poor, which cannot clearly demonstrate the results. It is good the recommended TNSRE and JFI can be included in the reference list. Please revise the paper carefully, so it can be reconsidered.

Round 3

Reviewer 2 Report

ok

Reviewer 4 Report

No more comments.